# Subcapsular Biloma following Endoscopic Retrograde Cholangiopancreatography and Endoscopic Biliary Sphincterotomy: A Case Report with a Mini Review of Literature

**DOI:** 10.3390/diagnostics13050831

**Published:** 2023-02-22

**Authors:** Natalia Valeria Pentara, Aristidis Ioannidis, Georgios Tzikos, Leonidas Kougias, Eleni Karlafti, Angeliki Chorti, Despoina Tsalkatidou, Antonios Michalopoulos, Daniel Paramythiotis

**Affiliations:** 1Department of Radiology, AHEPA General University Hospital, Aristotle University of Thessaloniki, 54636 Thessaloniki, Greece; 21st Propaedeutic Department of Surgery, AHEPA University Hospital, Aristotle University of Thessaloniki, 54636 Thessaloniki, Greece; 31st Propaedeutic Department of Internal Medicine, AHEPA Hospital, Aristotle University of Thessaloniki, 54636 Thessaloniki, Greece; 4Emergency Department, AHEPA University General Hospital, Aristotle University of Thessaloniki, 54636 Thessaloniki, Greece

**Keywords:** biloma, bile leakage, endoscopic retrograde cholangiopancreatography, percutaneous drainage

## Abstract

A biloma is a loculated, extrahepatic, intra-abdominal bile collection. It is an unusual condition with an incidence of 0.3–2% and is usually a result of choledocholithiasis, iatrogenic injury or abdominal trauma causing disruption to the biliary tree. Rarely, it will occur spontaneously, resulting in spontaneous bile leak. We herein present a rare case of biloma as a complication of endoscopic retrograde cholangiopancreatography (ERCP). A 54-year-old patient experienced right upper quadrant discomfort, following ERCP, endoscopic biliary sphincterotomy and stenting for choledocholithiasis. Initial abdominal ultrasound and computed tomography revealed an intrahepatic collection. Percutaneous aspiration under ultrasound guidance of yellow-green fluid confirmed the diagnosis, indicated infection and contributed to effective management. Most likely, a distal branch of the biliary tree was injured during the insertion of the guidewire through the common bile duct. Magnetic resonance image/magnetic resonance cholangiopancreatography contributed in the diagnosis of two seperate bilomas. Even though post ERCP biloma is an unusual complication, differential diagnosis of patients with right upper quadrant discomfort following an iatrogenic or traumatic event should always include biliary tree disruption. A combination of radiological imaging for diagnosis and minimal invasive technique to manage a biloma can prove to be successful.

## 1. Introduction

The term “biloma” was first introduced by Gould and Patel in 1979 in order to describe a loculated, encapsulated, extrahepatic biliary collection of bile [1]. However, the term was extended to include any intra-abdominal bile collection, external to the biliary tree and although many of them are encapsulated, the current definition does not require it to be as such [2]. Biloma is a rare condition with an incidence of 0.3% to 2.0% and is usually presented in patients aged 60 to 70 years old [3,4]. Biloma formation is most commonly a result of choledocholithiasis, iatrogenic injury and abdominal trauma, causing disruption to the biliary tree and furthermore bile leakage into the peritoneal cavity [4,5]. Although it is not common, bile leakage could also occur spontaneously, known as spontaneous bile leak (SBL), which is usually a diagnosis of exclusion [6]. ERCP is a combined endoscopic and fluoroscopic operation, incorporated with contrast material injection, allowing for radiologic imaging and if necessary therapeutic interventions. Some ERCP indications are obstructive jaundice, biliary or pancreatic ductal system condition treatment or biopsy, pancreatitis of unknown cause, nasobiliary drainage and biliary stenting among many others [7]. ERCP complications include post-ERCP pancreatitis (PEP) with a frequency of 3.5% which is the most common complication, and infections, such as cholangitis, gastrointestinal bleeding and duodenal or biliary perforations [8].

## 2. Case Report

Our patient, a 54-year-old female, had a history of chololithiasis after an ultrasound exam of the abdomen thirty years ago, due to non-specific abdominal pain, which revealed multiple echogenic shadowing stones within the gallbladder. During a recent visit to her personal physician, she was advised to have a magnetic resonance image (MRI) scan in order to evaluate the current state of her reported history of chololithiasis. MRI revealed multiple calculi in the gallbladder as well as within the common bile duct. The patient had a surgical history of right leg amputation due to an accident and was not taking any medication. The diagnosis was followed by ERCP. During this procedure, a guidewire under fluoroscopy was passed into the common bile duct, and contrast was injected, highlighting multiple filling defects (>4). Endoscopic papillary balloon dilation of the common bile duct was performed resulting in the gradual exit of bile duct stones. Subsequently, a plastic stent 10 fr–9 cm was placed in the common bile duct to ensure bile drainage (Figure 1). During the first day of her hospitalization, the patient started complaining of pain radiating to her back. Ultrasound examination of the abdomen revealed a collection of fluid with thickened hyperechoic walls compressing the liver parenchyma. The possibility of a tear of the biliary tree was considered. An abdomen computed tomography (CT) scan revealed a well-circumscribed subcapsular collection (5.7 × 3.5 cm) with an air–fluid level in segment VII of the liver, containing a contrast agent that was previously used in ERCP. It was firstly attributed to an intrahepatic biloma as a complication of the previously performed ERCP. Laboratory workup showed elevated white cell count (13,630 per microliter) and CRP at 231.3 mg/L. However, liver function tests (LFTs) remained within normal levels during the first two days: alanine transaminase(ALT) of 20.0 IU/L, aspartate transaminase (AST) of 32 IU/L, γ-GT of 10.0 IU/L, and bilirubin of 0.39 IU/L. Antibiotic treatment was initiated, Piperacillin/Tazobactam (4 + 0.5) g × 4 and Amikacin 375 mg × 2. Percutaneous drainage of the biloma was scheduled, as it was deemed necessary in order to fully manage this complication. On the third day of admission, a CT exam was repeated, revealing an increase in size of the forenamed subcapsular collection, as well as a rounded water-attenuation fluid collection with an air–fluid level in contact with the previous one. Differential diagnosis included an extension of the already existing collection or a completely different one (Figure 2). Percutaneous drainage of the biloma was performed under ultrasonography guidance. After the injection of 15 cc of lidocaine 1%, the collection was punctured through an intercostal approach with an 18 G Chiba needle. Aspiration of yellow-green fluid confirmed the diagnosis and the correct placement of the needle. An Amplatz superstiff guide wire was then inserted through the needle, and finally, an 8 F pigtail drainage catheter was placed in the collection. Injection of contrast was performed through the pigtail catheter, which opacified the subcapsular collection. Contrast material leak in the peritoneal cavity was not identified (Figure 3). Reduced drainage throughout the next days (ninth to twelfth day of admission), a repetitive CT scan and a magnetic resonance image/magnetic resonance cholangiopancreatography (MRI/MRCP) revealing minimal change in the size of the biloma strengthened the case of two separate bilomas (Figure 4). Percutaneous drainage was once more performed under ultrasound guidance, inserting a second 8 Fr pigtail drainage catheter, which passed through the first biloma and ended inside the second one. Contrast injection indicated no communication between the two bilomas (Figure 5). A sample of the drained fluid was collected, followed by laboratory analysis indicating E. coli infection. Follow-up CT showing both drainage catheters revealed capsular ring enhancement, designating inflammation and abscess formation. However a significant reduction in the size of the bilomas was detected (Figure 6). Over time, the amount of bile drainage decreased. The patient was discharged from the hospital after twenty-two days of hospitalization and came back seven days later in order to remove the two pigtails based on a follow-up CT, which revealed further size reduction of the bilomas and of the two pigtails’ drainage, which stopped. Thirty-eight days later, the patient came back in order to have a cholocystectomy. She had an uncomplicated postoperative course and was discharged from the hospital. Twenty days after her release, she underwent successful ERCP in order to remove the stent that was previously placed in the common bile duct and has been carefully monitored ever since.

## 3. Discussion

There are only some cases reported in the literature of this complication after ERCP [9,10,11]. A study by Enns et al. [12] of ERCP-related perforations showed that the incidence of ERCP-related perforations is less than 1%. Clinical presentation is variable, ranging from an incidental finding on imaging to right upper quadrant discomfort, abdominal fullness, nausea, vomiting, fever, jaundice, peritonitis without fever, or even severe biliary sepsis [2,3]. Reported causes of biloma formation following ERCP consider modalities of the endoscopic procedure, meaning the injection of contrast medium through the catheter inducing high pressure in proximal biliary ducts, the modification of the biliary epithelium by persistent cholangitis, or both [9]. In our case, the most likely scenario is that a distal branch of the biliary tree was injured by the tip of the guidewire. Its formation is thought to be via two mechanisms based on the pace of bile leakage. Secondary to slow leakage bile acids, which are known to have detergent and tissue destroying properties, cause mild inflammation in the neighboring abdominal tissues or liver parenchyma, prompting fibrosis and encapsulation. Additionally, rapid bile leaks could result in biliary peritonitis, where encapsulation may be present as a result of inflammatory adhesions [4]. Taking into account bilomas’ variable clinical presentation, radiological imaging was shown to be the foundation of diagnosis. Ultrasound exam, due to its non-invasiveness and rapid evaluation, is usually the primary medical imaging, revealing a wide range of findings from anechoic, well-circumscribed collections to large, complex fluid with multiple fine internal septa, but most frequently a cystic lesion [2,6]. Smaller bilomas can be missed, making abdominal CT the optimal method for its identification, usually showing a well-circumscribed, hypo-attenuated collection with clear margins and a density of less than 20 Hounsfield units [4,6,13]. Even though CT imaging has the benefit of providing us with useful information, such as the biloma’s location and surrounding structures, it is not the most effective diagnostic imaging modality to distinguish between differential diagnoses, such as postoperative hematoma, seroma, abscess, lymphocele, liver cyst or pseudocyst, making MR imaging and MRCP in some cases required [2,4]. A biloma typically produces a variable signal on T1-weighted images and high signal intensity on T2-weighted images, corresponding to the signal intensity of gallbladder fluid [14]. High T1 signal intensity and low T2 signal intensity material within the collection indicates concentrated bile layering [2]. MRI combined with MRCP provides convenient anatomical details of the intra- and extrahepatic bile ducts, sometimes identifying the location of the leakage [13]. Rim enhancement could be present as a consequence of reactive inflammation [4]. Hepatobiliary cholescintigraphy, which is a diagnostic nuclear medicine procedure using a radiotracer called Tc-99 m iminodiacetic acid to evaluate the biliary system, also known as HIDA scan, is considered an exceptionally effective radiological modality. Its profitability is based on the fact that the radiotracer follows the bilirubin metabolic pathway and is eventually excreted into the bile ducts, making it a beneficial tool in the diagnosis of gallbladder and biliary tree pathology [2,13,15]. The HIDA scan can reveal whether an active biliary leak is present and also helps to guide additional therapy [2]. Single positron emission computed tomography (SPECT) could also provide helpful details concerning the location of a possible active leak [2]. Invasive imaging techniques, such as ERCP and PTC, are useful where management is needed [2,13]. Imaging guided percutaneous transhepatic biliary drainage has both diagnostic and therapeutic roles by determining the proximal extent of biliary duct injury and providing fluid for laboratory analysis to confirm the diagnosis which is often required. In our case, percutaneous drainage of the biloma was performed under US guidance, preventing surgery. A study by Fatima et al. [16] concluded that most biliary perforations can be managed nonoperatively, and when operative treatment is required, the mortality rate increases. Nonetheless, further surgical management may be needed if percutaneous drainage is unsuccessful, multiloculated lesions are present or bile leaks are ongoing [4,6]. When it comes to blood testing, results vary from patients with no abnormalities to raised inflammatory markers and abnormal liver function tests. When a biloma is infected, blood cultures may unveil Gram-negative bacteremia. The most typical organisms found in the microbiological culture of biloma fluid are Enterobacteriaceae, followed by Enterococcus species [4].

## Figures and Tables

**Figure 1 diagnostics-13-00831-f001:**
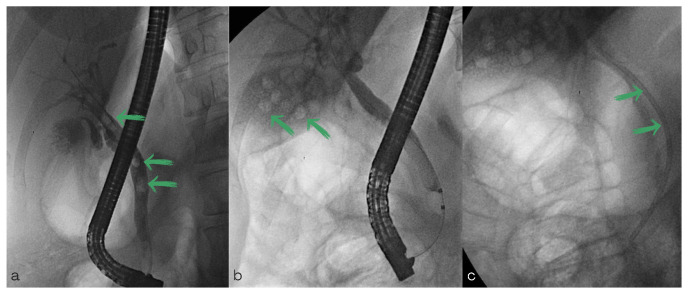
Endoscopic retrograde cholangiopancreatography fluoroscopy images. (**a**) Cannulation of the common bile duct and cholangiogram which shows intraluminal filling defects consistent with stones (green arrows). (**b**) Filling defects in the gallbladder (green arrows). (**c**) Biliary stent deployed (green arrows).

**Figure 2 diagnostics-13-00831-f002:**
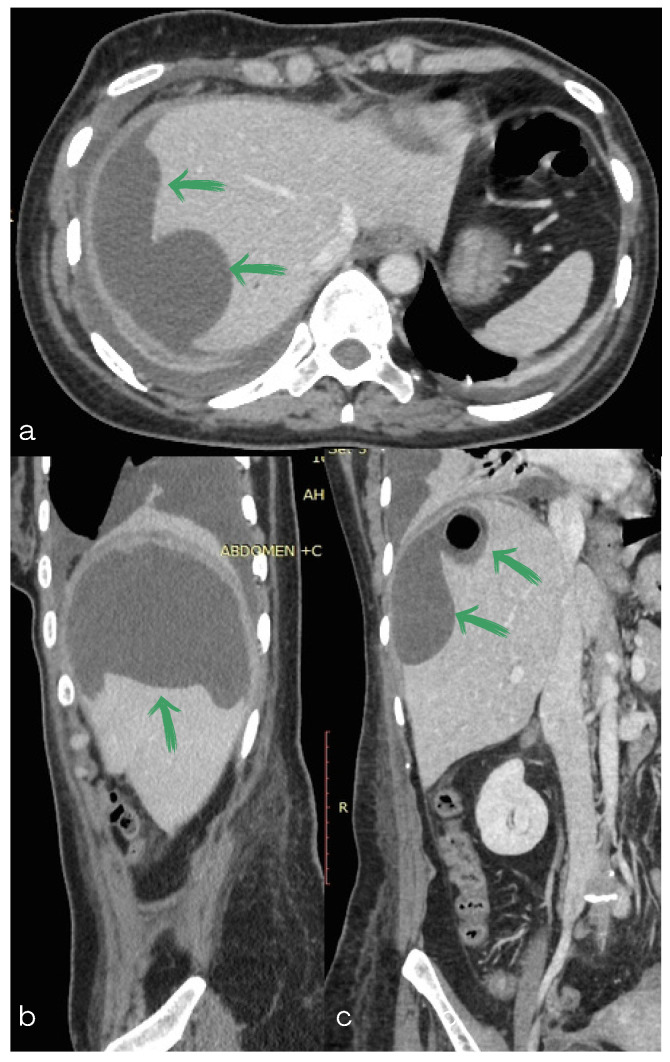
(**a**,**b**) Computed tomography scan of the abdomen: (**a**) axial, (**b**) sagittal and (**c**) coronal section showing a subcapsular collection as well as a rounded water-attenuation fluid collection with an air–fluid level in contact with the previous one (green arrows).

**Figure 3 diagnostics-13-00831-f003:**
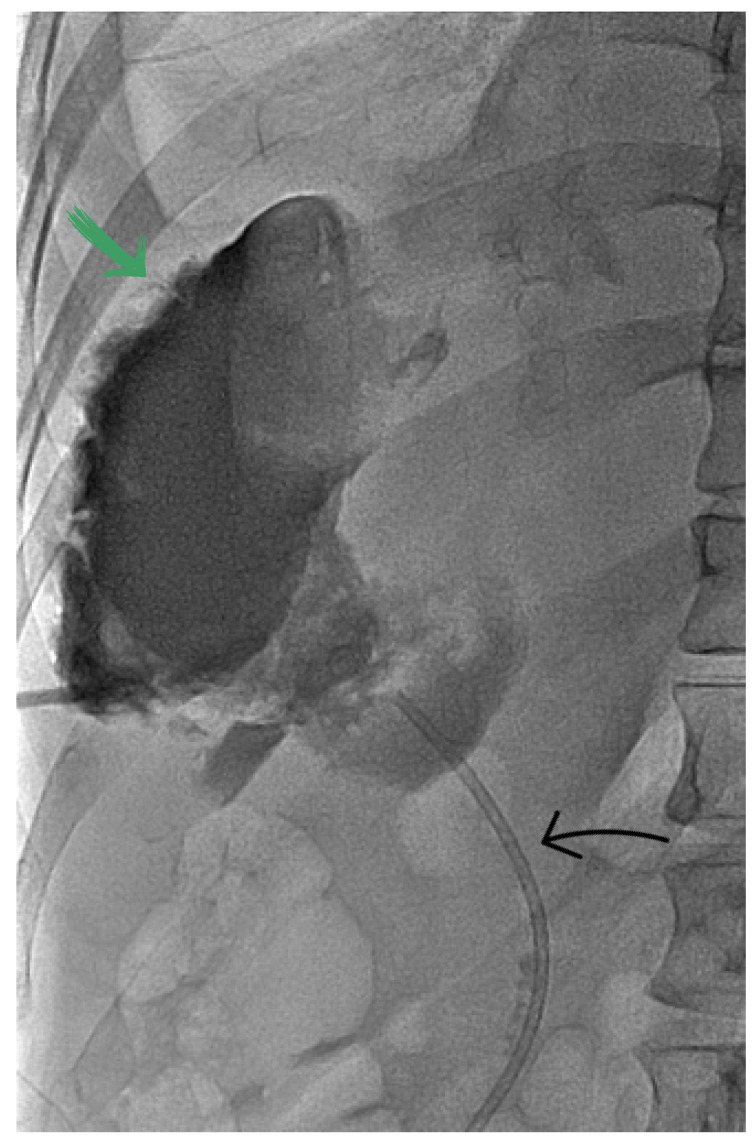
Contrast opacification of the subcapsular collection through the 8F pigtail catheter (green arrow). Previously placed stent in the common bile duct (black arrow).

**Figure 4 diagnostics-13-00831-f004:**
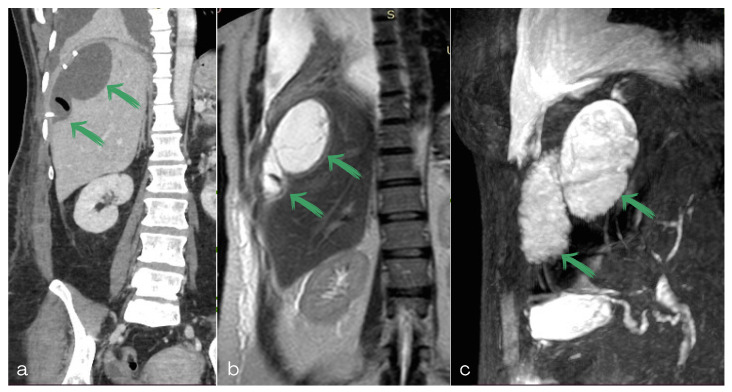
(**a**) Abdominal computed tomography scan, (**b**) magnetic resonance image/T2-weighted image of the abdomen and (**c**) magnetic resonance cholangiopancreatography of the abdomen revealing two separate bilomas.

**Figure 5 diagnostics-13-00831-f005:**
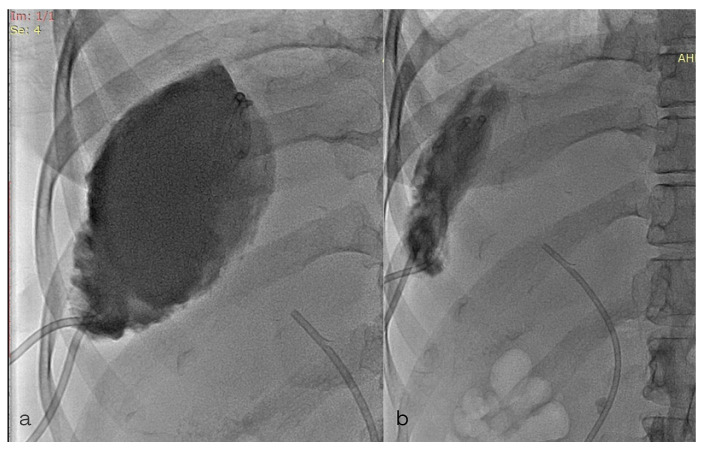
(**a**,**b**) Contrast opacification of the subcapsular collection through the 8F pigtail.

**Figure 6 diagnostics-13-00831-f006:**
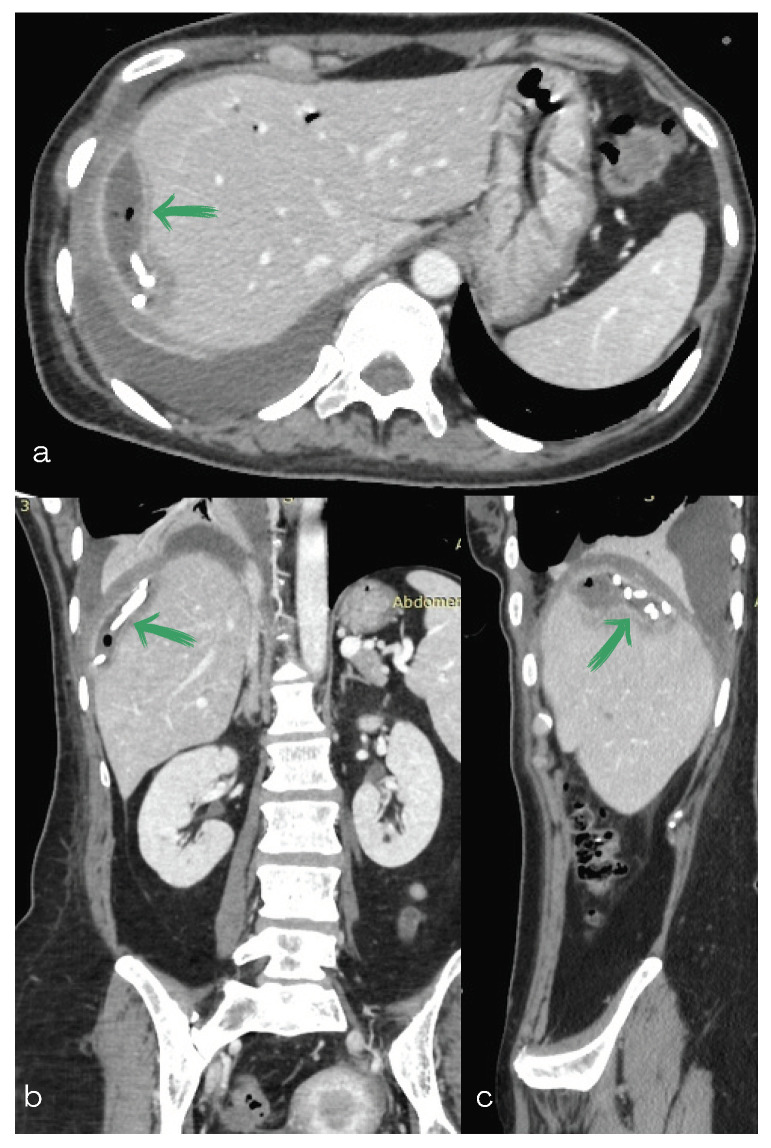
Computed tomography of the abdomen (**a**) axial (**b**) coronal and (**c**) sagittal section showing both pigtail drainage catheters as well as a capsular ring enhancement of the biloma (green arrow). Reduction in size of the two bilomas can also be seen.

## Data Availability

The data and materials/figures used in the current study are available from the corresponding author on reasonable request.

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
