# Peer review of "Subcapsular Biloma following Endoscopic Retrograde Cholangiopancreatography and Endoscopic Biliary Sphincterotomy: A Case Report with a Mini Review of Literature"

_diagnostics, 2023, doi:10.3390/diagnostics13050831_

Round 1
Reviewer 1 Report
The paper is far too long. Cut 1/2 of the introduction, case report and discussion. Remove conclusion. Remove either figure 1or 2. Remove figures 3&4
Author Response
We appreciate the time and effort that you dedicated to providing feedback on our manuscript and we are grateful for the insightful comments in order to improve our paper.
Response to Reviewer 1 Comments
Point 1: The paper is far too long. Cut 1/2 of the introduction, case report and discussion. Remove conclusion.
Response 1: I tried to reduce the size of the paper as much as possible while trying to maintain all the important information. Conclusion is removed.
Point 2: Remove either figure 1 or 2. Remove figures 3&4
Response 2: Figures 1,3 and 4 are removed.
Reviewer 2 Report
In this manuscript, the authors report on the case of a woman who, following ERCP performed for choledocolithiasis (associated to lithiasis of the gallbladder), developed a subglissonean S7 collection, consistent with double subcapsular biloma. These were successfully managed with antibiotics and percutaneous aspiration US guided and finally the patient could undergo cholecistectomy.
Subcapsular biloma following ERCP is a rare situation and its early diagnosis is important in order to avoid more severe complications related to an untreated biloma.
The manuscript is overall well written, the topic treated is adequately presented in the introduction, the clinical case is well described, and the discussion appropriately illustrates possbile mechanisms of biloma development after ERCP, as well as the diagnostic and treatment modalities available.
I have only few comments:
the biloma most likely developed between the liver parenchyma and the glissonean capsule: in this context, I believe that the term "sub-glissonean", instead of "subcapsular", better characterizes the presented clinical case.
The final management of the 2 pig tails used for the biloma drainage should be specified. When were they removed? Did the authors consider for removal the daily output volume? or the output characteristics?
Please specify.
Author Response
We appreciate the time and effort that you dedicated to providing feedback on our manuscript and we are grateful for the insightful comments in order to improve our paper.
Response to Reviewer 2 Comments
Point 1: the biloma most likely developed between the liver parenchyma and the glissonean capsule: in this context, I believe that the term "sub-glissonean", instead of "subcapsular", better characterizes the presented clinical case.
Response 1: The main terminology I found in literature was subcapsular biloma. However, I agree that sub-glissonean better characterizes the exact location of the biloma. If you agree and to be more specific we could use the term ''sub-glissonean capsule biloma''.
Point 2: The final management of the 2 pig tails used for the biloma drainage should be specified. When were they removed? Did the authors consider for removal the daily output volume? or the output characteristics?
Response 2: We added the following sentence in the case report part of the paper (line 84-86) explaining when and why the 2 pig tails used for the biloma where removed. '' The patient was discharged from the hospital after twenty two days of hospitalization and came back seven days later in order to remove the two pigtails based on a follow up CT which revealed further size reduction of the bilomas and on the two pigtails drainage which stopped''.
Round 2
Reviewer 1 Report
Accept as it is.